# Dysregulation of Microtubule Nucleating Proteins in Cancer Cells

**DOI:** 10.3390/cancers13225638

**Published:** 2021-11-11

**Authors:** Pavel Dráber, Eduarda Dráberová

**Affiliations:** Department of Biology of Cytoskeleton, Institute of Molecular Genetics, Czech Academy of Sciences, CZ-142 20 Prague, Czech Republic; drabere@img.cas.cz

**Keywords:** cancers, microtubule nucleation, γ-tubulin complexes

## Abstract

**Simple Summary:**

The dysfunction of microtubule nucleation in cancer cells changes the overall cytoskeleton organization and cellular physiology. This review focuses on the dysregulation of the γ-tubulin ring complex (γ-TuRC) proteins that are essential for microtubule nucleation. Recent research on the high-resolution structure of γ-TuRC has brought new insight into the microtubule nucleation mechanism. We discuss the effect of γ-TuRC protein overexpression on cancer cell behavior and new drugs directed to γ-tubulin that may offer a viable alternative to microtubule-targeting agents currently used in cancer chemotherapy.

**Abstract:**

In cells, microtubules typically nucleate from microtubule organizing centers, such as centrosomes. γ-Tubulin, which forms multiprotein complexes, is essential for nucleation. The γ-tubulin ring complex (γ-TuRC) is an efficient microtubule nucleator that requires additional centrosomal proteins for its activation and targeting. Evidence suggests that there is a dysfunction of centrosomal microtubule nucleation in cancer cells. Despite decades of molecular analysis of γ-TuRC and its interacting factors, the mechanisms of microtubule nucleation in normal and cancer cells remains obscure. Here, we review recent work on the high-resolution structure of γ-TuRC, which brings new insight into the mechanism of microtubule nucleation. We discuss the effects of γ-TuRC protein dysregulation on cancer cell behavior and new compounds targeting γ-tubulin. Drugs inhibiting γ-TuRC functions could represent an alternative to microtubule targeting agents in cancer chemotherapy.

## 1. Introduction

Microtubules are cytoskeletal polymers that are indispensable for vital cellular activities, such as cell division, migration, maintenance of cell shape, and ordered vesicle transport powered by motor proteins. They are also essential in organizing the spatial distribution of cellular organelles and signal transduction. Microtubules are intrinsically dynamic, as they oscillate stochastically between periods of growth and depolymerization in a process known as “dynamic instability of microtubules” [1]. Microtubules are assembled from globular αβ-tubulin heterodimers in a GTP-dependent manner to form a cylinder with an outer diameter of approximately 25 nm. The αβ-tubulin heterodimers are non-covalently joined in a head-to-tail fashion to form a linear protofilament. Thirteen parallel polar protofilaments associate laterally to form a left-handed helical microtubule wall. Microtubules are thus inherently polar and contain two structurally different ends: a slow-growing minus (−) end and a fast-growing plus (+) end. α-Tubulin is oriented toward the (−) end, while β-tubulin is oriented toward the (+) end of the microtubule [2]. Both the α- and β-tubulin subunits bind GTP; however, only the nucleotide on β-tubulin is hydrolyzed. GTP-bound tubulin dimers are incorporated into growing microtubules. GTP hydrolysis occurs with a delay after a GTP-tubulin dimer incorporates into the sheet-like structure of the growing microtubule tip. The growing microtubule ends thus maintain stabilizing GTP cap, the loss of which leads to rapid depolymerization [3]. In cells, microtubules are usually anchored to microtubule organizing centers (MTOCs) by their (−) ends, whereas the non-anchored (+) ends are highly dynamic. The dynamic properties of microtubules help to facilitate the remodeling of the microtubule network in response to internal or external signals. Although the structure of microtubules is conserved among various cell types, it can be adapted to highly divergent tasks by mechanisms that are still not fully understood. The incorporation of alternative tubulin isotypes, post-translational modifications (PTMs) of tubulin subunits, and interaction with microtubule-associated proteins (MAPs) regulate the microtubule properties. Intracellular microtubule organization is further controlled by the distribution of nucleation sites and proteins regulating microtubule organization [4].

Centrosomes, composed of two barrel-shaped and orthogonally arranged microtubule-based centrioles wrapped in a multiprotein matrix termed pericentriolar material (PCM), represent major MTOCs for nucleating microtubules in mammalian cells (Figure 1). Numerous scaffold and effector proteins, kinases, and phosphatases are involved in the formation of PCM-ordered layers and the organization of microtubules [5,6]. Centrosomes serve as hubs for the integration and coordination of various signaling pathways [7] and participate in actin filament organization [8,9]. Microtubule nucleation can also take place in other MTOC locations, such as the Golgi apparatus, nuclear envelope, plasma-membrane associated sites, pre-existing microtubules, and chromatin. These non-centrosomal sites play a significant role in the architecture of the microtubule network. γ-Tubulin, together with other proteins, named γ-tubulin complex proteins (GCPs), assembles into γ-tubulin ring complexes (γ-TuRCs), which represent the essential components for microtubule nucleation in various cellular locations [10,11,12,13].

In healthy cells, centrosome duplication ensures the formation of two functional centrosomes, which assemble bipolar spindles to avoid chromosomal aberrations in mitosis. By contrast, many cancer cells harbor extra centrosomes [14]. Centrosome amplification or its structural aberrations represent a hallmark of human cancers with direct consequences for chromosomal instability and can trigger cellular invasion [15,16]. Numerous studies have correlated the presence and the degree of centrosome amplification with indicators of poor prognoses, such as higher tumor grade and ability to recur and metastasize [17,18].

**Figure 1 cancers-13-05638-f001:**
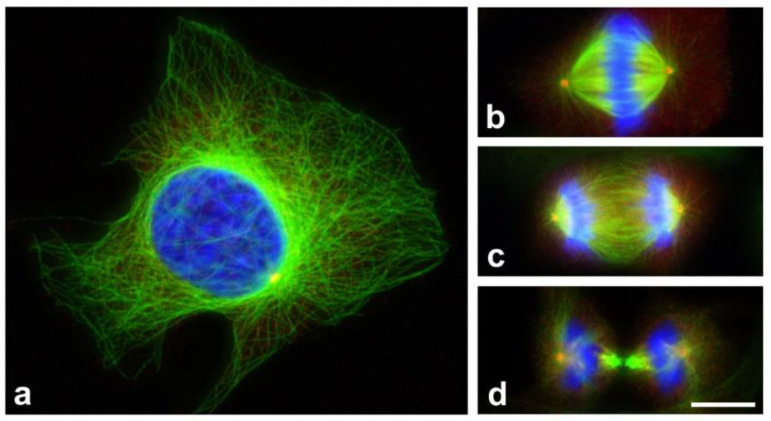
Microtubule nucleation during the cell cycle. Microtubules nucleated from centrosomes undergo substantial reorganization throughout the cell cycle. Human osteosarcoma U2OS cells in interphase (**a**), metaphase (**b**), anaphase (**c**), and late telophase (**d**) stained for microtubules with polyclonal anti-α-tubulin antibody (Genetex, GTX 15246; green) and for centrosomes with monoclonal anti-γ-tubulin antibody TU-30 [19] (red). DNA in blue. Formaldehyde-fixed samples were extracted with Triton X-100 and postfixed in methanol. Scale bar, 10 μm.

This review will focus on the latest research and emerging questions concerning centrosomal microtubule nucleation. Specific attention will be given to the high-resolution structure of γ-TuRC, dysregulation of γ-TuRC building proteins in cancer cells, and to new drugs targeting γ-tubulin functions.

## 2. Microtubules as Targets in Cancer Chemotherapy

Multiple genes encode both α-tubulin and β-tubulin isotypes, which differ in amino acid sequences mainly in their C-terminal ends. The C-terminal amino acids form unstructured tails that protrude from the microtubule surface. Differences among isotypes are often highly conserved in evolution, suggesting that they have functional significance [20]. Humans have nine isotypes for both α- and β-tubulin. Certain isotypes are ubiquitous, while others have cell- or tissue-specific expression and are important for the function of specialized microtubule arrays [21,22]. The amino acid sequences of tubulin isotypes incorporated into a microtubule and the PTMs on tubulin molecules generate the so-called tubulin code, which can lead to differences in microtubule properties. Well-characterized PTMs include acetylation, detyrosination, tyrosination, (poly)glutamylation, and (poly)glycylation. Additionally, there are many other PTMs, such as phosphorylation, methylation, polyamination, palmitoylation, arginylation, glycosylation, nitration, ubiquitylation, and sumoylation [23,24]. PTMs of tubulin isotypes result in multiple tubulin charge variants, or tubulin isoforms, which can be discriminated by high-resolution isoelectric focusing [25,26]. The tubulin code precisely controls microtubule properties and interactions of microtubules with a large number of MAPs. These can be divided into microtubule-stabilizing MAPs, microtubule-severing proteins, microtubule cross-linking proteins, and microtubule-assembly and disassembly promoters [4,27]. For example, acetylation makes microtubules more resistant to mechanical bending-induced breakage [28], and polyglutamylation regulates spastin-mediated microtubule severing [29].

Vertebrate β-tubulin isotypes have fairly distinct tissue distributions [30]. Compared with other β-tubulin isotypes, βIII-tubulin possesses distinctive biochemical properties [31]. Despite its restricted and predominantly neuronal cell-type distribution in normal tissues, the βIII isotype is expressed in a broad range of human tumors of neuronal and non-neuronal origin. At present, there is a general agreement that abnormal overexpression of βIII-tubulin in non-neuronal cancers is associated with an overall proclivity for aggressive tumor behavior and poor patient outcome [32].

Microtubule targeting agents (MTAs) are essential drug classes for cancer chemotherapy. Cancerous cells are characterized by, amongst other things, their potential to undergo continuous rounds of mitotic cell division. MTAs alter microtubule dynamics and interfere with the formation of the mitotic spindle. Errors in mitotic spindle function stimulate the spindle assembly checkpoint to block mitotic progression until all chromosomes are correctly attached by microtubules [33]. The long-term blocking of bipolar spindle formation by MTAs can eventually lead to apoptotic death, which is a strategy to combat cancer [34]. According to their effects on microtubules at high concentrations, MTAs are grouped into microtubule-stabilizing agents (MSAs) and microtubule destabilizing agents (MDAs). However, at low and clinically relevant concentrations, both MSAs and MDAs suppress microtubule dynamics without significantly affecting the amount of polymerized tubulin [35]. There are six known MTA binding sites identified on the tubulin dimer. The MSAs bind to taxane and laulimalide/peloruside sites, while MDAs bind to vinca, maytansine, colchicine, and pironetin sites. Except for the pironetin site, which is on α-tubulin, the other sites are located on β-tubulin [36]. The only MTAs approved by the Food and Drug Administration so far are those that bind to sites on β-tubulin.

The effectiveness of MTAs for cancer therapy is limited by their side effects and cancer cell drug resistance. The primary side effects are neurological and hematological and often limit the dose that can be administered, but several other side effects also occur during treatment with each individual drug [37]. Cancer cells also acquire resistance to MTAs through multiple mechanisms: (1) decreased cellular drug accumulation owing to the overexpression of membrane-bound drug efflux proteins, such as multidrug resistance-associated protein 1 and P-glycoprotein; (2) β-tubulin mutations at the drug binding sites; (3) altered expression of tubulin isotypes or MAPs; (4) changes to the microtubule dynamics induced by interactions with other cytoskeletal proteins; and (5) defects in apoptotic pathways [30]. Of the tubulin alterations, the increased abundance of βIII-tubulin is the most prevalent mechanism associated with the resistance to MTAs in patients [38].

## 3. γ-Tubulin and Microtubule Nucleation by γ-Tubulin Complexes

One of the key components required for microtubule nucleation and stabilization is γ-tubulin [39], a highly conserved albeit less abundant member of the tubulin superfamily. γ-Tubulin represents <1% of the total tubulin content in the cell [40]. In contrast to α- and β-tubulins, only two functional γ-tubulin genes (*TUBG1* and *TUBG2*) exist in humans, encoding two γ-tubulin isotypes. They are located on the 17th chromosome in tandem, and their coding sequences exhibit 94% sequence similarity. Although at the protein level, human γ-tubulin-1 and γ-tubulin-2 differ by only ten amino acids, they can be discriminated according to their electrophoretic and immunochemical properties [41,42]. Whereas γ-tubulin-1 is expressed ubiquitously, γ-tubulin-2 is primarily expressed in the brain [43,44]. Both γ-tubulins are nucleation competent [45]. γ-Tubulin binds GTP, which regulates its nucleation activity [46]. Mammalian γ-tubulins are phosphorylated at multiple sites [47], and protein kinases and phosphatases were repeatedly identified in γ-tubulin immunocomplexes, suggesting that phosphorylation might be involved in the regulation of γ-tubulin interactions [12]. Monoubiquitination of γ-tubulin inhibits microtubule nucleation [48], and γ-tubulin ubiquitination plays an important role in the degradation of γ-tubulin complexes [49].

γ-Tubulin, together with GCP2-6, forms a ~2.2-MDa γ-TuRC, which catalyzes microtubule nucleation in human cells, providing a structural template that mimics the microtubule geometry [50,51,52]. GCP2–6 each bind directly to γ-tubulin and assembles into a cone-shaped structure that templates microtubule assembly by binding αβ-tubulin dimers, and promoting their lateral interactions [53]. Two short homologous regions with low levels of sequence identity, the N-terminal GRIP (γ-tubulin ring protein) 1 domain and C-terminal GRIP2 domain, are unique to the GCPs. The GRIP domains are flexibly connected, allowing conformational rearrangement of γ-tubulins in the complex. γ-Tubulin interacts with GRIP2 domains, while GRIP1 domains create the primary interface between GCPs [53,54]. Recent high-resolution structural studies revealed details of the γ-TuRC structure [55,56,57,58]. In γ-TuRC, which has a width of ~30 nm and a height of ~25 nm, 14 γ-tubulin molecules are exposed at the open face of the cone and form a left-handed helix with a partial overlap of positions 1 and 14. The luminal bridge, a structural scaffold that lines the interior of the complex, is formed by two molecules of MZT (small mitotic spindle organizing protein) 1, while the protein MZT2 is located at the outer perimeter of γ-TuRC [59]. Surprisingly, actin was also shown to be an integral part of this internal structural scaffold. Actin binds with a high affinity to DNAseI. The treatment of purified γ-TuRC with DNaseI significantly inhibited their microtubule nucleation activity in vitro, and the pre-incubation of DNaseI with actin abolished this effect. Thus, actin is a bona fide structural component of γ-TuRC and has functional relevance in microtubule nucleation [56]. It appears that actin and associated luminal factors, together with MZT2, are jointly engaged in stabilizing the γ-TuRC spiral [55]. The structure and molecular architecture of the vertebrate γ-TuRC are shown in Figure 2. The configuration of γ-tubulins at the microtubule nucleation interface does not perfectly match the 13-fold symmetry of a microtubule. Spokes 1–8 in γ-TuRC, containing four GCP(2–3) units, termed γ-tubulin small complexes (γ-TuSC), follow microtubule symmetry. On the other hand, spokes 9–14, containing GCP4–6, introduce asymmetry in both the diameter and pitch of γ-TuRC and are not compatible with microtubule symmetry [60]. This may explain why cytosolic γ-TuRC has low nucleation activity [55] and conformational changes in the γ-TuRC structure are necessary for its activation [61]. Both activating protein factors and PTMs of γ-TuRC proteins can regulate this activation.

An important role in the induction of conformational changes is played by the activating factors CDK5RAP2 (cyclin-dependent kinase 5 regulatory subunit-associated protein 2) through its CM1 (centrosomin motif 1) domain [62] and NME7 (nucleoside diphosphate kinase 7) [63]. As γ-tubulin and GCPs are phosphorylated, their phosphorylation can modulate conformational changes that might be required for γ-TuRC activation [61]. Various associated proteins are additionally involved in the regulation of centrosomal microtubule nucleation. Targeting and anchoring proteins are not essential for the complex assembly but target γ-TuRCs to centrosomes and affect their recruitment. CDK5RAP2 is targeted to the centrosome through its CM2 (centrosomin motif 2) domain [64]. Once at the centrosome, the CM1 domain of CDK5RAP2 tethers the γ-TuRC. Sequential phosphorylation of NEDD1 (neural precursor cell-expressed developmentally down-regulated protein 1) [65] by Cdk1 and Plk1 kinases is required for targeting γ-TuRC to the centrosome [66]. Anchoring proteins AKAP9 (A-kinase anchor protein 9) [67], pericentrin [68], ninein [69], and Cep192 (centrosomal protein 192) [70] are also important for the localization of γ-TuRC to centrosomes. However, as these proteins are incorporated in PCM, they can also indirectly affect γ-TuRC binding. Additionally, modulating proteins are likewise vital for the regulation of centrosomal microtubule nucleation [12]. For example, TACC3 (transforming acidic coiled-coil-containing protein 3) has been found to stabilize γ-TuRC assembly from γ-TuSC [71,72]. The building components of γ-TuRC and its regulating proteins (activating, targeting, and anchoring) in the interphase centrosomes are summarized in Table 1. Proteins interacting with γ-TuRC can generate a heterogenous population of γ-TuRCs, which influences the recruitment of γ-TuRCs to MTOCs [73]. Although γ-TuRC is widely regarded as a bona fide PCM component nucleating microtubules, it also associates with the outer and inner walls of centriole and centriolar subdistal appendages [74].

It is increasingly evident that γ-tubulin has functions beyond microtubule nucleation. There have been repeated reports that mutations or deficiencies in γ-tubulin or GCPs alter (+)-end microtubule dynamics [75]. A microtubule nucleation-independent role of γ-tubulin complexes was also described for the control of the spindle assembly checkpoint (SAC) and mitotic exit [76] as well as for cell cycle progression in interphase [77]. There are also indications that nuclear γ-tubulin [78] modulates the activity of E2F transcription factors, which control cell cycle progression [79,80]. Although the functional significance is not yet clear, γ-tubulin associates with proteins involved in DNA damage checkpoints and DNA repair, such as Rad51 [81], BRCA1 [82], CDK5RAP3 [78], and ATR [83]. γ-Tubulin has an intrinsic property of generating oligomers in vitro [84] and was reported to be capable of forming fine fibers in cells [85,86]. The function of such fibrillar structures is unknown. It was suggested that they could serve scaffolding or sequestration functions and might also be involved in mechanotransduction, as they interact with the LINC (linker of the nucleoskeleton and cytoskeleton) complex [87,88]. γ-Tubulin has also been found in vesicles [89], recycling endosomes [90], and mitochondria [41,86].

## 4. Dysregulation of γ-Tubulin in Cancer Cells

Analysis of γ-tubulin expression in clinical tissue samples and cancer cell lines revealed the misregulation of this tubulin isotype, similar to the case of βIII-tubulin [91]. Multiple studies were performed on brain tumors and breast cancer, but changes in the γ-tubulin content were also reported in other cancers, as documented below.

Gliomas are the most prevalent group of central nervous system neoplasms, accounting for more than 70% of all brain tumors [92]. Gliomas are broadly classified as low-grade and high-grade gliomas. Glioblastoma multiforme (GBM) is the most malignant as well as the deadliest glioma and is refractory to currently available treatments. Oligonucleotide microarray analysis revealed that *TUBG1* transcripts were increased in GBM when compared with low-grade gliomas [93]. The immunohistochemical reactivity of γ-tubulin was significantly enhanced in high-grade anaplastic astrocytomas and GBM compared with low-grade diffuse astrocytomas [94]. Robust immunofluorescence staining was also found in glioblastoma cell lines (U87MG, U118MG, U138MG, and T98G). In contrast to pericentrin, which localized to centrosomes, γ-tubulin was found both on centrosomes and in the cytosol. Moreover, γ-tubulin localization in nuclei and nucleoli was clearly seen after acetic alcohol fixation [78]. A comparison of γ-tubulin expression and distribution in the primary culture of normal human astrocytes (NHA) and the human glioblastoma cell line T98G is shown in Figure 3. Interestingly, γ-tubulin clearly co-distributed with βIII-tubulin in GBM, while co-localization of both tubulins in low-grade gliomas was less evident. Both tubulins formed complexes in cytoplasmic pools from T98G cells [95]. mRNA levels for most centrosomal structural proteins and Aurora kinases were elevated in gliomas compared with normal human brain tissue. Significant differential expression between high- and low-grade gliomas was, however, detected only for γ-tubulin and Aurora A mRNAs, and upregulation of γ-tubulin was correlated with tumor grade [96,97]. Comparison of γ-tubulin expression at the transcript and protein level in glioblastoma cell lines and normal human astrocytes (NHA) confirmed enhanced expression of both γ-tubulin 1 and γ-tubulin 2 in glioblastoma cell lines U118MG, U138MG, and T98G [78]. These findings were further extended in the study, showing that the expression levels of γ-tubulin or Aurora A kinase were associated with patients’ age, astrocytoma grade, and patient survival and performance [98]. These results suggested that the aberrant expression of both βIII-tubulin and γ-tubulin may be linked to malignant changes in glial cells. This could be clinically useful for glioma staging and the development of novel treatment strategies [34]. Enhanced expression of γ-tubulin in brain tumors is not limited to gliomas. Overexpression of γ-tubulin was also reported in medulloblastomas and the medulloblastoma cell line D283Med. Overexpression of γ-tubulin was widespread in poorly differentiated, proliferating tumor cells stained for PCNA (proliferating cell nuclear antigen) but was significantly diminished in quiescent differentiating tumor cells undergoing neuritogenesis. Overexpression of γ-tubulin in the context of medulloblastomas may be a molecular signature of phenotypic dedifferentiation (anaplasia) and may be linked to tumor progression and worse clinical outcomes [99].

Breast cancer is the most frequent malignancy in women worldwide and is curable in ~70–80% of patients with early-stage, non-metastatic disease. Advanced breast cancer with distant organ metastases is considered incurable with currently available therapies [100]. Analysis of an extensive series of breast premalignant lesions and carcinoma revealed γ-tubulin gene amplification and concomitant protein overexpression in a significant fraction of atypical hyperplasia, in situ carcinomas, and invasive carcinomas [101,102]. In a panel of human breast cancer cell lines, prominent cytosolic and nuclear γ-tubulin immunostaining was found in aggressive breast cancer cell lines. By contrast, γ-tubulin localization remained largely centrosomal in non-tumorigenic cell lines. It was suggested that changes in γ-tubulin solubility are regulated on the PTM level [103]. An essential role in regulating microtubule nucleation in breast cancer is played by γ-tubulin ubiquitylation by the BRCA1 (breast cancer type 1 susceptibility protein)/BARD1 (BRCA1-associated RING domain protein 1) E3 ligase complex. The ubiquitination of γ-tubulin results in its detachment from the centrosome and inhibition of microtubule nucleation. The loss of the BRCA1 protein, as occurs in many breast cancers, induces centrosome amplification and enhances the formation of large microtubule asters from centrosomes [48,104,105]. A significant role in the regulation of breast cancer centrosomes is also played by deubiquitylase BAP1 (BRCA1-associated protein-1). The removal of ubiquitin from γ-tubulin by BAP1 induces the recruitment of unmodified γ-tubulin to the centrosome during mitosis [106].

Elevated expression of γ-tubulin was reported in a large cohort of clinical tissue samples from non-small cell lung cancer (NSCLC). While in the non-neoplastic cells of the airway epithelium in situ, γ-tubulin exhibited predominantly apical surface and pericentriolar localizations, markedly increased cytosolic γ-tubulin immunoreactivity was detected in clinical tumor specimens. A notable increase of γ-tubulin expression was found in stage III compared with lower-stage tumors (stages I/II). Increased γ-tubulin expression was also found in NSCLC tumor cell lines NCI-H460 and NCI-H69 compared with small airway epithelial cells (SAEC) [107,108]. Additionally, increased γ-tubulin expression was reported in surgically resected laryngeal carcinomas [109], thyroid carcinomas [110], ovarian cancer [111], and prostate cancer cells [112].

The increased expression of γ-tubulin in some cancer types may be linked to increased microtubule nucleating capacity through conventional or supernumerary centrosomes, which could result in enhanced invasiveness [113]. The increased level of γ-tubulin might also be connected to abnormal centrosomal functions due to the structural changes in centrioles [16]. Moreover, non-centrosomal microtubule nucleation could also be affected. As γ-tubulin has functions beyond microtubule nucleation, its overexpression can change microtubule (+)-end dynamics [75] as well as intracellular signaling pathways because γ-tubulin interacts with various protein kinases [12]. Dysregulation of γ-tubulin in cancer cells can also modulate cell cycle [77], transcription [79,80], DNA damage checkpoint [78], and DNA repair [81,83].

## 5. Dysregulation of the Other γ-TuRC Building Components

Contrary to γ-tubulin, data on changes in the expression of GCPs in cancer cells are limited, and dysregulation was reported only for GCP2 and GCP3 [114]. In accordance with increased expression of γ-tubulin in glioma cells, the expression of GCP2/GCP3 was augmented both at the transcript and protein level in U87MG, U118MG, U138MG, T98G, and KNS glioma cell lines compared with NHA. Both proteins were concentrated on the centrosomes of interphase cells, but they were also located in the cytosol and nuclei/nucleoli. GCP2 and GCP3 were found in complexes with γ-tubulin in the nucleoli, as confirmed by reciprocal immunoprecipitation experiments and immunoelectron microscopy. GCP2 and GCP3 depletion caused the accumulation of cells in G2/M and mitotic delay but did not affect nucleolar integrity. Similarly, as in the case of γ-tubulin overexpression [78], the enhanced level of GCP2 antagonized the inhibitory effect of the CDK5RAP3 (cyclin-dependent kinase 5 regulatory subunit-associated protein 3) on DNA damage G2/M checkpoint activity. The immunohistochemical reactivity for γ-tubulin was significantly enhanced in GBM compared with low-grade diffuse gliomas. These findings suggest that γ-TuSC protein dysregulation in glioblastomas may be linked to altered transcriptional checkpoint activity or interaction with signaling pathways associated with a malignant phenotype [114]. GCP2 was shown to be upregulated at the transcript level in the taxol-resistant ovarian cancer cell line SKOV3 and was described to be associated with sensitization of the NSCLC cell line NCI-H1155 to taxol [115].

Concerning MZT proteins, which are integral parts of γ-TuRCs, increased expression was recently reported for MZT2A in NSCLC. Both MZT2A mRNA and protein were upregulated in NSCLC tissues, which correlated with larger NSCLC size, lymph node metastasis, and poor NSCLC prognosis. MZT2A was highly expressed in all tested NSCLC cell lines except for NCI-H460 compared with normal bronchial cells. MZT2A overexpression promoted NSCLC cell viability and invasion. Moreover, MZT2A indirectly induced Akt phosphorylation to promote NSCLC proliferation and invasion [116].

Actin plays essential roles in aberrant processes in cancer, including signaling, morphology, motility, and division. As actin has functional relevance in microtubule nucleation [56], changes in the actin level in cancer cell types might also affect microtubule nucleation. Mass spectrometry analysis suggested that preferentially β- or γ-actin is integrated into γ-TuRC, but the nearly identical sequences between actin isoforms indicate that the isotype of γ-TuRC-associated actin may depend on the expression profiles in particular cells [117]. The upregulation of β-actin or γ-actin was reported in many cancers [118]. Moreover, γ-actin was shown to be involved in the regulation of centrosome function and mitotic progression in cancer cells [119]. Interestingly, profilin 1, a principal control component of actin polymerization, also modulates microtubule dynamics [120,121]. It was reported recently that profilin 1 interacts with γ-TuRCs and attenuates centrosomal microtubule nucleation [122]. Profilin has multiple isoforms, and it has been shown that the ratio of profilin 1 to profilin 2 transcripts in primary tumors decreases. Elevated levels of profilin 1 correlated with tumor-suppressive effects [123]. It was proposed that in the moving cells, profilin 1 and profilin 1-actin are recruited to the advancing cell edge to support upregulation of actin turnover and the formation of lamellipodia. Consequently, profilin concentration at the centrosome is lowered, which causes increased nucleation of microtubules from the centrosome [124]. Accordingly, profilin 1 might modulate both cell motility and centrosomal microtubule nucleation in highly invasive cancer cells.

## 6. Therapeutic Potential of γ-Tubulin Targeting

The clinical usefulness of many MTAs has been hampered by cancer cell drug resistance. Several studies have suggested that γ-tubulin might be a good candidate for the development of new anticancer drugs. First, γ-tubulin accumulates on the centrosome at the onset of mitosis to facilitate bipolar spindle assembly [125]. Second, elevated microtubule nucleation from amplified centrosomes enhances the invasiveness of tumor cells [15]. Third, centrosomal microtubule nucleation has been shown to be an attractive drug target [126,127]. Finally, γ-tubulin is overexpressed in different cancers, as presented in Section 4 of this review. Reversing the high nucleation capacity of centrosomes through γ-tubulin inhibition may reduce the aggressiveness and metastatic potential of the wide range of cancer cells with supernumerary centrosomes.

It was reported that colchicine and combretastatin A, which bind to colchicine binding site on β-tubulin, also associate with recombinant human γ-tubulin 1 [128]. Because γ-tubulin is structurally quite similar to β-tubulin [129], several laboratories started to develop γ-tubulin-specific inhibitors from known drugs which bind to a colchicine-binding site on β-tubulin. This approach identified gatastatin (O^7^-benzyl glaziovianin A derivative) as an inhibitor that had a higher affinity toward γ-tubulin in comparison with αβ-tubulin. Gatastatin blocked GTP binding to γ-tubulin, inhibited centrosomal microtubule nucleation, and impaired spindle formation [130]. Molecular docking and molecular dynamics methods revealed the binding of gatastatin to the GTP binding site on γ-tubulin [131]. However, the cytotoxicity of gatastatin to cancer cells was relatively weak compared with that of conventional MTAs, such as paclitaxel or vinblastine. On the other hand, the cytotoxicity of gatastatin was significantly increased by Plk1 inhibitor BI 2536. Dual inhibition of γ-tubulin and Plk1 arrested cell cycle progression, generated abnormal spindles, and caused spindle assembly checkpoint-dependent mitotic cell death by impairing centrosome functions. These results raised the possibility of Plk1 and γ-tubulin inhibitor co-treatment as novel cancer chemotherapy [132]. Further optimization of gatastatin recently led to the preparation of the highly specific γ-tubulin inhibitor gatastatin 2 (O^6^ -propargyl-gatastain), which showed potent cytotoxicity, induced abnormal spindle formation with misaligned chromosomes, and inhibition of microtubule nucleation [133].

The same approach of chemical modification of known drugs binding to the β-tubulin colchicine site was used in the case of resveratrol (3,5,4′-trihydroxy-trans-stilbene, a naturally occurring phenol), which binds to the colchicine site and has anti-carcinogenic activity. The synthetic manipulation of the stilbene scaffold of resveratrol led to new analogs with improved anticancer activity and better bioavailability. 3,4,4′-Trimethoxystilbene (3,4,4′-TMS) inhibited cell proliferation, depolymerized the mitotic spindle, and produced a fragmentation of the pericentriolar material. Computer-assisted docking studies showed the interaction of 3,4,4′-TMS with γ-tubulin [134]. Noscapine (phthalide isoquinoline alkaloid) is a non-addictive opioid with antitumor activity that easily transverses the blood–brain barrier and does not induce peripheral neuropathy, which is common with other MTAs. It was shown that 9′-bromonoscapine, which has significant anticancer activity, binds at or near the β-tubulin colchicine site [135]. Molecular docking, molecular dynamics simulation, and binding free energy calculation indicated its binding into a pocket located at the binding interface between two adjacent γ-tubulin molecules [136]. Structurally simplified noscapine analogs that selectively bind only to γ-tubulin could represent new tools for the treatment of glioblastoma with a high level of γ-tubulin [137].

Finally, the targeting of nuclear-specific γ-tubulin function(s) was also reported. γ-Tubulin and RB1 (retinoblastoma transcriptional corepressor 1) moderate each other’s expression by direct binding to sites for E2Fs transcription factors on the *TUBG1* and *RB1* promoter regions. The RB1 tumor suppressor pathway is well-described, and it is highly mutated in a large spectrum of cancers [138]. In the absence of nuclear γ-tubulin and RB1, the uncontrolled transcriptional activity of E2Fs upregulates apoptotic genes, causing cell death [139]. It was shown that terpenoid citral dimethyl acetal (CDA) increased E2F activity without affecting microtubules and showed a cytotoxic effect in cells with a nonfunctional RB1 pathway. In silico and in vitro experiments demonstrated that CDA prevents GTP binding to γ-tubulin. It was proposed that drugs specifically designed to inhibit the nuclear activity of γ-tubulin could target malignant tumors without affecting healthy cells [140].

## 7. Conclusions and Perspectives

More than 25 years of research on γ-tubulin complexes led to the conclusion that they are almost universally involved in microtubule nucleation. The γ-TuRCs have been shown to form microtubule templates, and their attachment to both centrosomal and non-centrosomal sites correlates with an increase in microtubule nucleating activity. The recent structural studies on γ-TuRC have been highly illustrative, but a central open question for the future will be to understand how the various proteins involved in the regulation of microtubule nucleation control the conformation and activity of γ-TuRCs. Little is known about the upstream signaling pathways ensuring that these proteins initiate microtubule nucleation at the correct location and time. The importance of centrosomal kinases and phosphatases in the regulation of nucleation is emerging, but the details are only partially understood. A thorough comprehension of microtubule nucleation should clarify the relevance of γ-TuRC dysregulation in cancer cells.

Understanding the role of γ-tubulin isotypes in DNA damage checkpoints and signaling pathways to DNA repair will be essential to elucidate the role of γ-tubulin in tumorigenesis. Although centrosome amplification and increases in γ-tubulin cell content are present in various human tumors, extensive clinical randomized studies are necessary to further evaluate the predictive value of the γ-tubulin level in the context of different clinical stages, histological types, tumor grades, and treatment settings. It is also becoming increasingly clear that microtubules and actin filaments work together in core cellular processes, and their dynamic properties are often intertwined. It remains to be determined how the dysregulation of microtubule nucleating and regulatory proteins affects the cross-talk between these two cytoskeletal systems in highly motile aggressive cancer cells.

Currently, the most common and most effective chemotherapy compounds are MTAs that bind microtubules directly. Targeting γ-TuRCs may offer a viable alternative to perturbing cancer cells. A future challenge will be to develop drugs that can inhibit γ-TuRC function in a precise manner. The elucidation of the three-dimensional structure of γ-tubulin and γ-TuRC has provided an opportunity for rational drug design aimed at generating compounds that will target microtubule nucleation in more therapeutically efficacious ways compared with the presently available MTAs. In this respect, non-essential γ-TuRC proteins could also represent potential targets for anticancer drugs, as their inhibition may affect only subsets of γ-TuRCs.

## Figures and Tables

**Figure 2 cancers-13-05638-f002:**
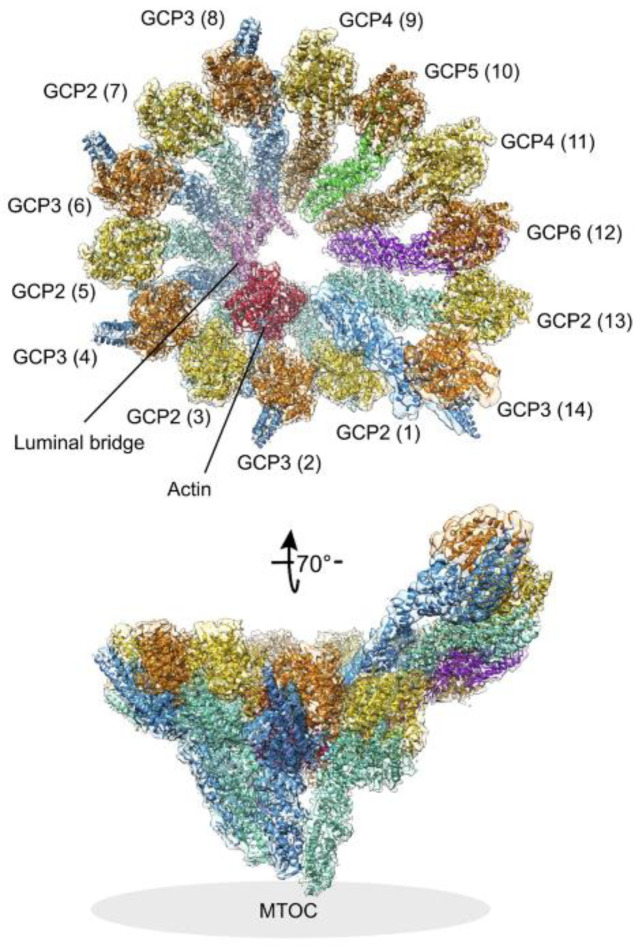
Structure and molecular architecture of the γ-TuRC. The general architecture of the vertebrate left-handed γ-TuRC spiral as determined by cryo-EM single-particle analysis. γ-Tubulins (yellow, orange), GCP2 (aquamarine), GCP3 (blue), GCP4 (brown), GCP5 (green), GCP6 (purple), actin (red), and the luminal bridge (pink) are shown. The spoke numbers are given in brackets. In the tilted view, the approximate location of the MTOC is indicated. (Reprinted by permission from Elsevier: (Curr. Opin. Struct. Biol.; Zupa et al. [60]), copyright (2021).

**Figure 3 cancers-13-05638-f003:**
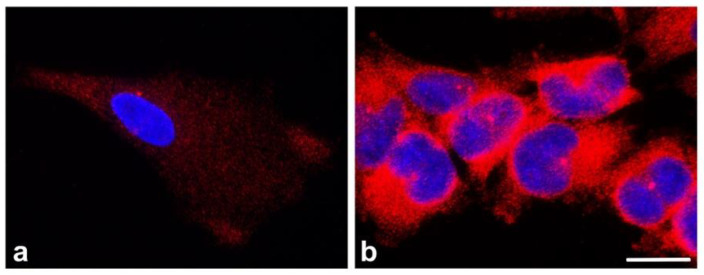
Differential expression and localization of γ-tubulin in human astrocytes and glioblastoma cells. Comparison of γ-tubulin distribution (red) in primary culture of human astrocytes (Lonza CC-2565) (**a**) and human glioblastoma cell line T98G (**b**) using monoclonal antibody TU-30 [19]. Nuclei in blue. Note the abnormal accumulation of γ-tubulin outside centrosomes in glioblastoma cells. Fluorescence images were captured and processed in precisely the same manner. Formaldehyde-fixed samples were extracted with Triton X-100 and postfixed in methanol. Scale bar, 20 μm.

**Table 1 cancers-13-05638-t001:** Building components of γ-TuRC and regulatory proteins of centrosomal microtubule nucleation in interphase cells.

γ-TuRC Proteins	Activating Proteins	Targeting Proteins	Anchoring Proteins
γ-tubulin	CDK5RAP2	CDK5RAP2	AKAP9
GCP2	NME7	NEDD1	Cep192
GCP3			Ninein
GCP4			Pericentrin
GCP5			
GCP6			
MZT1			
MZT2			
Actin			

Protein classification based on recent reviews [12,60].

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
