# Peer review of "Dysregulation of Microtubule Nucleating Proteins in Cancer Cells"

_cancers, 2021, doi:10.3390/cancers13225638_

Round 1

Reviewer 1 Report

The manuscript by Draber and Draberova reviews the current knowledge on y-TURC complexes and their potential as targets in cancer therapies. The authors start by introducing microtubules and they describe the microtubule-targeting agents currently used in cancer therapy and their limits. They then introduce the gamma-tubulin ring complexes that initiate microtubule nucleation, review their misregulation in the context of cancer and discuss their potential use as targets for anticancer drugs. They present an interesting overview in a clearly written and comprehensive way. They provide substantial background and report is up to date. I would recommend acceptance pending some minor stylistic and clarification modifications.

  • In their introduction, the authors state that “g-tubulin complexes represent the essential components for MT nucleation in various cellular locations” (lane 59-61). I think the authors should be more precise and explain what they mean by g-tubulin complexes. I guess that they refer to the y-TURCs but those are introduced only later in their text.
  • Lane 52 « centrosomes … represent major MTOCs for nucleation microtubules » should be “for microtubule nucleation” or “for nucleating microtubules”.
  • Legend Figure 1: It would be nice to state more precisely which antibodies have been used, especially because I wonder which polyclonal anti-tubulin has been used, since to my knowledge most anti-tubulin antibodies nicely staining microtubules by immunofluorescence are monoclonal.
  • Lane 133: gamma-tubulin at end of line uses wrong character.
  • In Chapter 3, the authors may consider adding a few lines about TACC3 as a regulator of y-TURC function. Indeed, TACC3 has been found to stabilize the interaction between y-TURC and y-TUSC (Singh et al. JBC 2014, Rajeev et al. BMC Mol Cell Biol 2019) and was shown to be misregulated in many cancer tissues (Yao et al. Oncogene 2014; for review Ding et al. Cytoskeleton 2017).

Reviewer 2 Report

In this review, Draber and Draberova first describe the current literature on microtubule nucleation and the evidence for α- and β-tubulin isotypes dysregulation in cancers. They then describe the role of the γ-TuRC in MT nucleation, including recent advances on the high-resolution structure of the γ-TuRC. The authors then describe how γ-tubulin is also dysregulated in certain cancers and describe how the new knowledge of the γ-TuRC structure opens up new avenues of research on anticancer drugs targeting γ-tubulin and the γ-TuRC.

This review is interesting, very detailed and well structured.

Broad comments:

It would be useful to give references to original publications rather than to another review, where the reader will then have to trace the references to get to the original data. Examples of this would be in line 314 (ref 9).

Some further details on the role of the GTPase activity of the tubulins and how this affects their function / structure would be nice.

Specific comments:

Line 16: reword - “For nucleation is essential γ-tubulin” should read “γ-tubulin is essential for nucleation”.

Line 18: reword – “Accumulating evidence suggests” perhaps to something like “evidence suggests there is a dysfunction of...”

Line 30: reword – “maintenance of cell shape, division, migration, and ordered vesicle transport” should be reordered for clarity - for example “cell division, migration, maintenance of cell shape, and ordered vesicle transport”.

Line 40: reword – “exposing” is the wrong word in this context here.

Line 42-43: reword - “Microtubule dynamics help remodel microtubules in response to internal or external signals.” Do the authors mean that “This dynamic property of microtubules helps to facilitate the remodelling of the MT network in response to internal or external signals”?

Line 44: Perhaps adding a short description of conservation of Tubulin and/or microtubules through evolution would be nice here.

Line 52: reword/typo – “represent major MTOCs for nucleation microtubules in mammalian cells” could be changed to “represent major MTOCs for nucleating microtubules in mammalian cells”.

Line 80: reword/typo – “Both α- and β-tubulin are encoded by multiple genes making tubulin isotypes”. Do the authors mean “Multiple genes encode both a-tubulin isotypes and b-tubulin isotypes, which differ in amino acid sequences mainly in their C-terminal ends.”

Line 81: Where does C-terminal end of tubulin lie in relation to the MT itself? Do these C-termini point out / or into the lumen of the microtubule.  It might be nice to give a tiny bit more detail here.

Line 85-87: Clarify maybe a bit more what is meant by the “tubulin code”. Do the authors mean that the sequence of Tubulin isotypes incorporated into a MT and the subsequent PTM on Tubulin molecules leads to a ‘tubulin code’, and that this so-called ‘tubulin code’ that can lead to differences in MT properties? Would it be possible to give an example?

Line 91: typo/unclear meaning – “Such diversification of tubulin isotypes results in multiple tubulin charge variants, tubulin isoforms”. Not clear exactly the meaning of this sentence.

Line 104: The statement “Cancerous cells are characterized by their potential to undergo continuous rounds of mitotic cell division” could be rephrased slightly to acknowledge that cancerous cells are also characterised by other properties (and indeed non-cancerous cells – e.g. stem cells have the potential to undergo continuous rounds of mitotic cell division). This could be re-phrase to something like “Cancerous cells are characterized by, amongst other things, their potential to undergo continuous rounds of mitotic cell division”.

Line 113: Maybe the authors could clarify what they mean by “microtubule polymer mass”. Does the suppression microtubule dynamics lead to a different number, or density of microtubules?

Line 117: reword/clarify - “Until now, all MTAs approved by the Food and Drug Administration are those binding to β-tubulin.” Do the authors mean that “The only MTAs approved by the Food and Drug Administration so far are those binding to β-tubulin” or do they mean that “recently MTAs binding to both a-tubulin and b-tubulin have now been approved by the FDA”?

Line 120: reword/clarify – “Neurological and haematological side effects are the primary and often dose-limiting toxicities in the treated patients”. Do the authors mean that “The primary side effects are neurological and haematological and often limit the dose that can be administered”?

Line 133: reword/clarify - “Minor” is ambiguous – do the authors mean “less abundant”?

Line 133: Typo “y-Tubulin” should read “γ-Tubulin”.

Line 187: Perhaps it might be nice to give a reference for how the activation is regulated.

Line 196: Perhaps the statement “γ-TuRC centrosome targeting is mediated by CDK5RAP2 through its CM2 (centrosomin motif 2) domain” is a little misleading. The CM2 domain is required to target CDK5RAP2 to the centrosome. Once at the centrosome the CM1 domain of CDK5RAP2 then provides a receptor for targeting the γ-TuRC to the centrosome. This sentence needs altering to clarify this.

Line 213: reword – “Increasing pieces of evidence indicate that γ-tubulin has functions beyond microtubule nucleation” - Perhaps to something like “It is increasingly evident that that γ-tubulin has functions beyond microtubule nucleation.”

Line 230: As the sentence “Analysis of γ-tubulin expression in clinical tissue samples and cancer cell lines revealed dysregulation of this tubulin isotype” refers to the regulation of γ-tubulin gene expression should this perhaps be “Analysis of γ-tubulin expression in clinical tissue samples and cancer cell lines revealed mis-regulation of this tubulin isotype”.

Line 232: Perhaps it would be nice to give a reference for these multiple studies.

Line 362: reword – Maybe it would be nice to use a different phrase other than “ held back” which is rather colloquial.

Line 369: Rather than “chapter 4” this should read perhaps “section 4”.

Author Response

This manuscript is a resubmission of an earlier submission. The following is a list of the peer review reports and author responses from that submission.